# COMPRESSION AND ACCELERATION OF DEEP NEURAL NETWORKS: A VECTOR QUANTIZATION APPROACH

## ABSTRACT

In the advancing field of deep learning, we witness the emergence of models that are getting larger, with an increasing number of parameters. However, this progress carries a downside, as it requires more powerful hardware, thereby restricting the utilization of deep learning models, particularly on edge devices. Hence, a vital requirement arises for compressing and accelerating deep learning models to enable their widespread deployment. Majority of recent studies proposed compression or acceleration based on pruning, low-precision quantization, matrix factorization and knowledge distillation. In this paper, we present a novel paradigm for compressing and accelerating deep learning models by harnessing vector quantization, a widely-recognized method in data compression. Our technique directly applies vector quantization to the neural network weights. More precisely, a VQ-DNN model divides weight parameters into equally sized segments, with the values of these segments exclusively derived from a compact codebook of values. During training, a VQ-DNN model learns both the codebook values and the mapping to model weight parameters. Our work demonstrates that vector quantization leads to more efficient implementations of matrix multiplications and convolution operations, ultimately reducing the computational cost. This efficiency enables us to accelerate and compress a wide range of models, including both Convolutional Neural Networks (CNNs) and vision transformers. We present experimental results on datasets such as CIFAR-10, ImageNet, and EuroSat using popular architectures like VGG16, ResNet, and ViT models. In all scenarios, VQ-DNN reduces model size by over 95%, surpassing state-of-the-art methods. Furthermore, it achieves comparable or superior reductions in Floating Point Operations (FLOPs) compared to existing methods, contingent on the dataset and model configuration.

## 1 INTRODUCTION

In recent years, deep learning has transformed the AI field, capturing the public's attention with its impressive performance (Sarker, 2021). However, as the field progresses, models are growing in size and complexity. These larger models require more advanced hardware, which hinders the widespread adoption of deep learning, especially on edge devices (Wei et al., 2022; Fang et al., 2023). In many practical scenarios, a modest reduction in accuracy is tolerable if it results in substantial compression and acceleration.

From the early days of DNNs, compression and acceleration has been an active research area (Le-Cun et al., 1989; Whitley et al., 1990; Castellano et al., 1997). Low-rank decomposition have been proposed to factorize a convolutional layer into several smaller layers (Rakhuba et al., 2014; Astrid & Lee, 2017; Zhang et al., 2015). Network pruning has also been a very popular approach where a large number of components of a pre-trained model is removed to achieve compression and acceleration. These components can be weight parameters (Han et al., 2015) entire neurons (Srinivas & Babu, 2015), convolution filters (Li et al., 2016), an entire layer (Chen & Zhao, 2018), intermediate activations (Georgiadis, 2019), etc. Some approaches designed a new model architecture that are inherently more compact and efficient, such as MobileNet (Howard et al., 2017) and MSDNET (Huang et al., 2017). Some approaches are proposed to quantize the network by storing and operating on a fewer bit parameters of 16 bits (Oh et al., 2018) or down to even 1-bits (i.e. binarization) (Hubara et al., 2016).

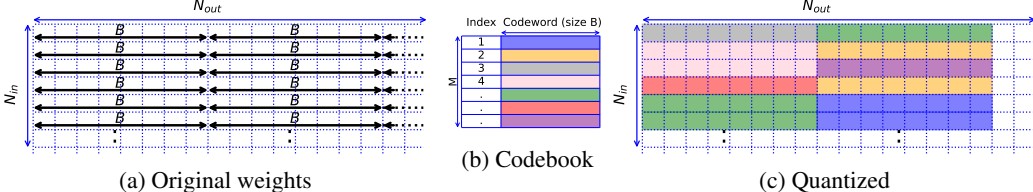

Figure 1: **a**: The weight matrix of a standard linear layer, with dimensions $N_{in} \times N_{out}$, undergoes segmentation into uniform-size blocks B row-wise. Within each block, values are substituted with the nearest codeword from a shared global codebook, which is shared across all linear layers. **b**: A codebook comprising M codewords, each having a size of B.**c**: After vector quantization (VQ), values within each block in the weight matrix are replaced with an entry from the codebook. The visualization uses distinct colors to represent entries within the codebook. The resulting weight matrix functions similarly to a conventional linear layer in a neural network.

In this paper, we introduce a groundbreaking approach to compressing and accelerating deep learning models through the utilization of Vector Quantization (VQ). VQ, a data compression technique commonly applied in signal processing and data compression, serves to represent a collection of data points (vectors) with a limited set of representative vectors. Our research addresses a fundamental question arising from prior work: Is it feasible to portray the model parameters using a limited set of representative vectors? Drawing inspiration from the neural discrete representation learning approach by (Van Den Oord et al., 2017), we propose an innovative method in which a codebook is learned alongside the model, forcing the model to exclusively rely on weight parameters derived from this codebook.

In essence, our approach induces a model to use numerous repetitive blocks (segments) of parameters obtained from a learned codebook (Fig. 1), due to the codebook size being considerably smaller than the model parameters. This approach yields two key advantages:

- **Reduced Storage Requirement**: By representing model parameters using just a few bits that indicate their index to the codebook, we significantly reduce the storage demands of the model.

- **Optimized Computational Efficiency**: Since many parameters within each layer become duplicates, their computation over the input data produces identical results. In Section 3.4, we will illustrate how this redundancy can be leveraged to minimize the overall number of Floating-Point Operations (FLOPs) required at inference.

In this work, we introduce vector quantized deep neural networks. We first demonstrate how matrix multiplication and convolution process can be vector quantized. Hence, it allows us to vector quantize most common architectures. We perform experiments on various models, including VGG, ResNet, and ViT, and on several datasets, including CIFAR-10, ImageNet, and EuroSat. Our findings demonstrate that these models can be effectively vector quantized using as few as 16 codebook entries, incurring only a minor accuracy penalty. Furthermore, by holding the codebook fixed during training, we delve into the adaptability of these models. In a notable experiment, we successfully transfer a codebook trained for one task, ImageNet, to an entirely distinct task, EuroSat (see Appendix A.6). Furthermore, by using GradCAM++ (Chattopadhay et al., 2018), we show that a vector quantized model learns similar features to a conventional model (see Appendix A.7)

## 2 RELATED WORK

Most previous works on model compression or acceleration can be divided into four categories: pruning, low-precision weights, knowledge distillation, and matrix decomposition. Pruning can be done on weight parameters (Han et al., 2015) entire neurons (Srinivas & Babu, 2015), convolution filters (Li et al., 2016), an entire layer (Chen & Zhao, 2018), intermediate activations (Georgiadis, 2019), etc. However, some of these approaches, such as weight pruning and neuron pruning, lead to an unstructured model which is hard to implement efficiently. Most pruning methods are post-training and often remove unimportant parameters iteratively. There are approaches that utilize the

data to filter out unimportant filters (Lin et al., 2020b; Wang et al., 2018; Molchanov et al., 2019). Data-free approaches often use $l_p$-norms of a weight/filter as proxy for importance (He et al., 2018; Ye et al., 2018; Zhuo et al., 2018). There are also some works that utilize regularization techniques to induce sparsity (Yang et al., 2019a; Wang et al., 2020a; Zhuang et al., 2020).

Knowledge distillation improves the performance of a student model by taking the soft-labels from a teacher model (Hinton et al., 2015). If the student model is smaller, it can be construed as a compression method, as suggested in (Wang et al., 2019; Li et al., 2020; Duong et al., 2019). Matrix decomposition is another technique representing the weight matrix as a low-rank product of two smaller matrices by using decomposition techniques, such as SVD (Denton et al., 2014; Alvarez & Salzmann, 2017) or CP-decomposition (Lebedev et al., 2014). While matrix decomposition successfully compress and accelerate DNNs, it often results in large accuracy penalty, particularly under high compression settings (Lin et al., 2020a).

Leveraging low-precision parameters, including 16-bit floats, 8-bit integers, and even the binarization of parameters is another approach for the compression and acceleration of models (Jacob et al., 2018; Lee & Nirjon, 2019; Hubara et al., 2016; Yang et al., 2019b; Umuroglu et al., 2017).

Several approaches share a conceptual similarity with our method, although they differ in their underlying inspiration, specifications, and implementation. (Gong et al., 2014) demonstrated the practicality of clustering layer parameters, both individually and within sub-matrices. Building upon this, (Wu et al., 2016) and its subsequent extension (Cheng et al., 2017) further evolved the clustering idea to achieve both network compression and computational acceleration. However, two primary distinctions set their approach apart from ours: Firstly, training the codebook alongside the model enables us to regulate the entire model to employ only a single codebook, enhancing compression. Secondly, as the previous work is a post-training method, the replacement of weights with codebook values incurs more significant accuracy degradation, particularly if no re-training is done. The most related paper to our approach, as described in (Minsik et al., 2022), addresses the second drawback of previous methods by suggesting a differentiable k-means clustering layer that incorporates an attention mechanism during training. However, they continue to use separate clusters for each layer. Additionally, their primary focus was on model compression, and they did not tackle computational acceleration. For a more detailed comparison with this work, please refer to Appendix A.8.

# 3 VQ-DNN

Vector quantization (VQ) compresses data by representing it with a reduced set of representative vectors known as codewords. Initially, a codebook is assigned by using either random values or, more frequently, clustering algorithms. Then, each data point is replaced with its closest codeword.

In 2017, (Van Den Oord et al., 2017) introduced vector quantization to deep neural networks for quantizing VAE's latent representation. When presented with an activation value segment $x$, the authors forwarded the nearest codeword $z(x)$ to the decoder in the forward pass, instead. During the backward pass, the gradient $\nabla_z \mathcal{L}$ remained unchanged and was applied to $x$. Using the same method, we apply vector quantization to the network's weights instead of activations. During the forward pass, we quantize the weights of the layers and perform layer operations, and during the backward pass, we simply apply the unaltered gradients to the weights.

Many modern deep learning architectures incorporate Fully Connected (i.e., Linear) layers and 2D convolution layers, often employing 1x1 or 3x3 filters. These layers typically account for a significant portion of the architecture's parameters. In this paper, we focus exclusively on these two layer types, although a similar approach could extend to other neural network layers. In this section, we unveil the details of how we apply vector quantization to both linear and 2D convolutional layers. Additionally, we introduce an efficient implementation method capable of significantly compressing and accelerating Deep Neural Networks (DNNs).

## 3.1 Vector Quantized Linear Layer

A linear layer contains a weight matrix $W \in \mathbb{R}^{N_{in} \times N_{out}}$. $N_{in}$ and $N_{out}$ represent the dimensions of the input and output for that layer, respectively. Here we break down this weight matrix $W$ into blocks of size $B$. Consequently, a VQ-linear layer requires two components: an index matrix

$I \in \mathbb{Z}^{N_{in}/B \times N_{out}}$ and a codebook $C \in \mathbb{R}^{M \times B}$. Here, $M$ and $B$ are hyper-parameters determining the number of entries in the codebook and the size of each block, respectively. Note that each $B$ consecutive parameters is mapped to a single entry of the codebook. In our analysis, we assume that the indices can be stored using just a single byte, although it's worth noting that in the majority of our experiments, even fewer bits have proven sufficient. It's important to highlight that, in this paper, we do not vector quantize the bias parameters since they are relatively small in comparison with the weight parameters. Assuming a precision of 4 bytes, a linear layer requires $4 \times N_{in} \times N_{out}$ bytes of storage. In contrast, a vector-quantized linear layer demands $4 \times M \times B$ for the codebook and $(N_{in}/B) \times N_{out}$ for the index matrix. We demonstrate that a single codebook can be shared among different layers of the same type, and hence, all those layers only require a single, compact, full-precision codebook. This results in a remarkably compressed model.

## 3.2 Vector Quantized 2D Convolution Layer

A 2D convolution layer consists of a weight matrix, denoted as $W \in \mathbb{R}^{C_{out} \times C_{in} \times k \times k}$, where $C_{in}$ and $C_{out}$ represent the number of input and output channels, respectively, and $k$ indicates the filter size. In our experiments, we focus on quantizing 3x3 and 1x1 2D convolution layers. Unlike the quantization approach used for linear layers, here we break down the weight matrix $W$ into $B$ consecutive $k \times k$ 2D filters rather than individual parameters. In other words, each entry in the convolution codebook has dimensions of $B \times k \times k$. Consequently, a VQ-Conv2D layer necessitates an index matrix $I \in \mathbb{Z}^{C_{out} \times C_{in}/B}$ and a codebook $C \in \mathbb{R}^{M \times B \times K^2}$, where $M$ represents the number of entries in the codebook. Assuming that one byte is sufficient to store each codebook index, the total storage required is $C_{in}/B \times C_{out} + 4 \times M \times B \times k^2$ bytes. As an example, for a convolutional layer with input and output channel sizes of 64, and a codebook with 16 entries of size $4 \times 3 \times 3$, the total number of parameters is reduced from $64 \times 64 \times 3 \times 3 = 36,864$ to $64 \times (64/4) + 16 \times 4 \times 3 \times 3 = 1,600$. This compression is especially significant considering that the codebook can be shared among multiple layers, resulting in an exceptionally compact model.

## 3.3 Training

To train a vector-quantized network, we maintain the original weight parameters $W$ in place. During the forward pass, we replace each block of $B$ parameters (or filters) with the nearest codebook entry and then perform the layer's operation (i.e matrix multiplication or convolution) using the quantized weights $W_q$. In the backward pass, we need to update both the weight parameters $W$ and the codebook entries. Updating $W$ is straightforward; we simply use the gradients obtained for $W_q$. However, updating the codebook entries can be approached in two ways: 1. Using a loss term to minimize the distance between codebook entries and the parameter blocks that match them during the forward pass. 2. Using a closed-form solution, which is equivalent to the average of the blocks of parameters that match each codebook entry during the forward pass.

Following a similar approach to VQ-VAE (Van Den Oord et al., 2017), we also incorporate another loss term to encourage weight parameters to stay close to the values in the codebook. This helps prevent extreme values in weight parameters that could destabilize training. For the first approach, the total loss is given by Eq. 1:

$$\sum_{i=1}^{n} y_i . \log(\hat{y}_i) + \sum_{j \in \text{VQ-layers}} \sum_{b} (||\text{sg}[W_{jb}] - e||_2^2 + ||W_{jb} - \text{sg}[e]||_2^2) \tag{1}$$

Here, $W_{jb}$ represents a block of parameters $b$ in layer $j$, and $e$ is an entry in the codebook that was mapped to $W_{jb}$ during the forward pass. In this equation, *sg* represents the stop gradient operator, which enforces its operand to remain non-updated by the optimizer. The second term moves the codebook entries to align with the mapped weight parameters, while the third term moves the weight parameters to correspond to their associated codebook entries. For the second approach, the total loss is as Eq. 2:

$$\sum_{i=1}^{n} y_i . \log(\hat{y}_i) + \sum_{j \in \text{VQ-layers}} \sum_{b} (||W_{jb} - e||_2^2) \tag{2}$$

In this case, the codebook entries are updated using exponential moving averages, and they are not updated via the optimizer. Therefore, we omit the stop gradient operator here. For more detailed implementation information regarding the second approach, please refer to Appendix A of the work by (Van Den Oord et al., 2017). In our experiments, we observed that the second approach not only converges more quickly but also yields slightly superior results.

## 3.4 ACCELERATION

After training a vector-quantized model, we can replace each block of weight parameters with the closest codeword from the codebook and subsequently remove the original weights. The resulting model is ready for direct use. However, due to the limited number of entries in the codebook, typically around 16, each codebook entry repeatedly appears within each layer's weight matrix. This leads to a significant number of redundant multiplications and convolution operations. To optimize computation efficiency, we perform these operations just once and reuse the results throughout, thereby reducing the number of Floating Point Operations (FLOPs).

Let's delve into the details of how this acceleration technique works: For a vector-quantized linear layer, we begin by dividing the input, denoted as $Inp \in \mathbb{R}^{N_{in}}$, into $N_{in}/B$ blocks, each of size $B$. Subsequently, we perform a multiplication of each block with the entire codebook. The outcome is referred to as the "multiplication lookup matrix", which encompasses all the necessary multiplication operations for a given layer. Since the final results require summation along the input dimension, we can sum each consecutive set of $B$ values within the multiplication lookup matrix to avoid redundant summations at a later stage. Consequently, the size of the multiplication lookup matrix becomes $N_{in}/B \times M$. To compute the output of the layer, we utilize the index matrix to look up the multiplication results from the multiplication lookup matrix and perform the final summation along the input dimension.

In a standard linear layer, the computational cost is proportional to $N_{in} \times N_{out}$ FLOPs, as documented in prior work (Wu et al., 2016; Cheng et al., 2017). In the case of a vector quantized linear layer, we initially require $(N_{in}/B) \times M \times B$ FLOPs to construct a multiplication lookup table. Subsequently, an additional $N_{in}/B \times N_{out}$ FLOPs are needed to compute the output. It's worth noting that a larger value of B results in a reduction in the number of FLOPs required. To illustrate this, consider a layer with 1024 inputs and 1024 outputs, using a codebook with 16 entries, each of size 8. This reduces the FLOPs from the standard $1,024 \times 1,024 = 1,048,576$ to just $1024 \times 16 + (1,024/8) \times 1,024 = 147,456$.

It's also important to highlight that we treat the matrix multiplication involving weight parameters for the query, key, and value in the multihead attention layers as a linear layer. This treatment results in a significant reduction in the overall number of multiplication operations. A conventional convolution layer typically requires a substantial number of FLOPs, calculated as $H_{out} \times W_{out} \times C_{in} \times C_{out} \times k^2$ (Wu et al., 2016; Li et al., 2016). In contrast, a vector quantized convolution layer follows a two-step process akin to that of a vector quantized linear layer. In the first step, we perform a convolution operation using each entry from the codebook across $B$ consecutive input channels, summing the results. This process generates a convolution lookup matrix with dimensions $H_{out} \times W_{out} \times C_{in}/L \times C_{out}$, requiring a total of $H_{in} \times W_{in} \times C_{in} \times M \times k^2$ FLOPs. In the second step, we construct the final output by extracting values from the convolution lookup matrix using layer's indices. This step incurs an additional cost of $H_{out} \times W_{out} \times C_{in}/L \times C_{out}$ FLOPs. Notably, the second operation does not include the $k^2$ term. Consequently, for convolutions with $k > 1$, we can reduce the number of FLOPs, even when we don't reduce the number of input channels (i.e., when $L = 1$).

## 4 EXPERIMENTS

In this section, we perform a thorough evaluation of VQ-DNNs using various models and datasets. We also compare their effectiveness against several state-of-the-art methods. It's essential to highlight that most methods in the literature are post-training techniques, intended for meticulously trained models that achieve the highest accuracy. In contrast, VQ-DNN is designed to be integrated into the training process. Training a deep learning architecture to reach its highest accuracy is a significant challenge, often requiring an extensive process of adjusting settings and applying various

techniques to achieve good results. In some situations, especially when dealing with large models like vision transformers and large datasets like ImageNet, limitations in computational power prevent us from trying out many different experiments to get the best possible outcome. Therefore, we chose a specific training recipe that gives us satisfactory results and used the same recipe with minor changes for all our experiments. To make sure we're comparing results fairly, we trained an equivalent non-vector quantized (non-VQ) deep neural network (DNN) model using the same recipe as a reference point. As a result, we report the accuracy drop compared to the non-VQ DNN model we trained. For more details on the training recipe refer to Appendix A.1. The models we evaluate encompass VGG16, ResNet18, ResNet50, and ViT_b_16, and we evaluate them across datasets including CIFAR-10, ImageNet (ILSVRC2012), and EuroSat.

Table 1: Training results for vector-quantized ResNet50 on CIFAR-10 using various codebook sizes

| K1/K3 codebook (M, B) | Acc. (%) | △ Acc. (%) | FLOPs | FLOPs ↓ (%) | Params | Params ↓ (%) |
|---|---|---|---|---|---|---|
| Original ResNet | 94.15% | - | 1304.69M | - | 23.52M | - |
| (16,8)/(32,8) | 93.08% | ↓ 1.07 | 327.82M | 74.87% | 0.50M | 97.89% |
| (16,8)/(32,4) | 93.53% | ↓ 0.62 | 336.20M | 74.23% | 0.56M | 97.61% |
| (16,8)/(32,2) | 93.30% | ↓ 0.85 | 352.98M | 72.95% | 0.63M | 97.33% |
| (16,8)/(32,1) | **93.87%** | ↓ **0.28** | 386.54M | 70.37% | 0.77M | 96.73% |
| (16,8)/(24,8) | 93.10% | ↓ 1.05 | 282.99M | 78.31% | 0.50M | 97.89% |
| (16,8)/(24,4) | 93.29% | ↓ 0.86 | 291.38M | 77.67% | 0.53M | 97.73% |
| (16,8)/(24,2) | 93.23% | ↓ 0.92 | 308.15M | 76.38% | 0.61M | 97.40% |
| (16,8)/(24,1) | 93.28% | ↓ 0.87 | 341.71M | 73.81% | 0.77M | 96.73% |
| (16,8)/(16,8) | 92.48% | ↓ 1.67 | 238.16M | **81.75%** | 0.52M | 97.78% |
| (16,8)/(16,4) | 93.03% | ↓ 1.12 | 246.55M | 81.10% | 0.53M | 97.73% |
| (16,8)/(16,2) | 93.17% | ↓ 0.98 | 263.33M | 79.82% | 0.61M | 97.40% |
| (16,8)/(16,1) | 93.08% | ↓ 1.07 | 296.88M | 77.25% | 0.78M | 96.70% |
| (24,8)/(24,8) | 92.73% | ↓ 1.42 | 305.47M | 76.59% | 0.50M | 97.89% |
| (24,8)/(24,4) | 92.93% | ↓ 1.22 | 313.86M | 75.94% | 0.53M | 97.73% |
| (24,8)/(24,2) | 93.12% | ↓ 1.03 | 330.63M | 74.66% | 0.61M | 97.40% |
| (24,8)/(24,1) | 93.57% | ↓ 0.58 | 364.19M | 72.09% | 0.77M | 96.73% |
| (24,16)/(24,8) | 92.47% | ↓ 1.68 | 262.21M | 79.90% | **0.31M** | **98.70%** |
| (24,16)/(24,4) | 92.94% | ↓ 1.21 | 270.60M | 79.26% | 0.34M | 98.53% |
| (24,16)/(24,2) | 92.90% | ↓ 1.25 | 287.38M | 77.97% | 0.42M | 98.20% |
| (24,16)/(24,1) | 93.45% | ↓ 0.70 | 320.93M | 75.40% | 0.58M | 97.53% |
| (16,16)/(24,8) | 92.02% | ↓ 2.13 | 239.73M | 81.63% | **0.31M** | **98.70%** |
| (16,16)/(24,4) | 92.36% | ↓ 1.79 | 248.12M | 80.98% | 0.34M | 98.54% |
| (16,16)/(24,2) | 92.77% | ↓ 1.38 | 264.90M | 79.70% | 0.42M | 98.20% |
| (16,16)/(24,1) | 93.09% | ↓ 1.06 | 298.46M | 77.12% | 0.58M | 97.54% |
| (24,8)/(32,2) | 93.58% | ↓ 0.57 | 375.46M | 71.22% | 0.61M | 97.40% |
| (24,8)/(32,1) | 93.46% | ↓ 0.69 | 409.01M | 68.65% | 0.77M | 96.73% |

## 4.1 CIFAR-10

**ResNet18 and ResNet50**: Majority of the parameters of resnet family architectures are in $3 \times 3$ and $1 \times 1$ Conv2D layers. Hence, we use only two codebooks here, one for 2D convolutions of $1 \times 1$ size, referred to as **k1**, and one for $3 \times 3$, referred to as **k3**. The final linear layer of the architecture remain un-quantized as it contains a small number of parameters. $M$ and $B$ hyper-parameters are shown in $(M, B)$ tuples. Note that for k3 codebooks each entry actually has a size of $B \times 3 \times 3$. For that reason, even if we set $B = 1$, we still replace 9 full precision parameters with a single byte index. Results for resnet-50 is shown in Table 1. Our method can reduce $25.6M$ parameters to less than $800$ thousand. Increasing $M$, the number of entries in the codebook, can slightly improve the accuracy while it does not increase the number of parameters significantly. Decreasing $M$, on the other hand, can improve both FLOPs and model parameters significantly at the cost of accuracy. For resnet-18, results are presented in Table 2. It is noteworthy that the majority of our models demonstrate a level of efficiency surpassing that of representing each full-precision parameter, typically requiring 4 bytes, with merely a single bit. To elaborate, this implies that, from a technical perspective, we utilize less than a single bit for the storage of each parameter.

**Comparison with the state of the art**: In this section, we have performed the training of multiple vector-quantized ResNet50 models and subsequently compared them with state-of-the-art compression and acceleration methods. In each model group, we select a codebook size for ResNet50 such that the number of Floating-Point Operations (FLOPs) stays within a small range. The outcomes, as detailed in Table 3, reveal a significant edge for VQ-DNNs in terms of compression when compared to state-of-the-art methods. Concerning accuracy, VQ-DNNs maintain their competitiveness, with

Table 2: Training results for vector-quantized ResNet18 on CIFAR-10 using various codebook sizes.

| K1/K3 codebook (M, B) | Acc. (%) | △ Acc. (%) | FLOPs | FLOPs ↓ (%) | Params | Params ↓ (%) |
|---|---|---|---|---|---|---|
| Original ResNet | 93.68% | - | 556.65M | - | 11.17M | - |
| (32,8)/(32,8) | 89.07% | ↓ 4.61 | 173.13M | 68.90% | 0.11M | 99.03% |
| (32,8)/(32,4) | 91.00% | ↓ 2.68 | 180.74M | 67.53% | 0.12M | 98.90% |
| (32,8)/(32,2) | 92.33% | ↓ 1.35 | 195.94M | 64.80% | 0.19M | 98.32% |
| (32,8)/(32,1) | 92.98% | ↓ 0.70 | 226.35M | 59.34% | 0.33M | 97.01% |
| (16,8)/(16,8) | 87.72% | ↓ 5.96 | **92.26M** | **83.43%** | **0.08M** | **99.25%** |
| (16,8)/(16,4) | 89.85% | ↓ 3.83 | 99.87M | 82.06% | 0.11M | 99.01% |
| (16,8)/(16,2) | 90.73% | ↓ 2.95 | 115.07M | 79.33% | 0.18M | 98.38% |
| (16,8)/(16,1) | 92.23% | ↓ 1.45 | 145.48M | 73.87% | 0.33M | 97.04% |
| (32,8)/(96,1) | 93.28% | ↓ 0.40 | 542.50M | 2.54% | 0.35M | 96.90% |
| (16,8)/(96,1) | 93.23% | ↓ 0.45 | 540.66M | 2.87% | 0.35M | 96.91% |
| (16,8)/(64,1) | **93.29%** | **↓ 0.39** | 382.59M | 31.27% | 0.34M | 96.96% |
| (16,8)/(32,1) | 92.99% | ↓ 0.69 | 224.52M | 59.67% | 0.33M | 97.01% |

accuracy degradation remaining below the $1\%$ threshold, even though some alternative methods may outperform us in terms of accuracy.

Table 3: Comparisons of different methods for compressing/accelerating ResNet-50 on CIFAR-10 with different pruning ratios. Baseline accuracy of our model is $94.15\%$ while other approaches used a well-trained model with baseline accuracy of $95.22\%$.

| ID | Acc. (%) | △ Acc. (%) | FLOPs | FLOPs ↓ (%) | Params ↓ (%) |
|---|---|---|---|---|---|
| FPGM (He et al., 2019) | 94.73% | ↓ 0.49 | 420M | 67.8% | 65.9% |
| SG-CNN (Guo et al., 2020) | 95.12% | ↓ 0.10 | 415M | 68.2% | 66.1% |
| KPGP (Zhang et al., 2022) | **95.23%** | **↑ 0.01** | 420M | 67.8% | 65.9% |
| VQ k1(16,8)/k3(32,1) | 93.87% | ↓ 0.28 | 386.5M | 70.3% | 96.7% |
| FPGM (He et al., 2019) | 93.59% | ↓ 1.63 | 273M | 79.0% | 76.9% |
| SG-CNN (Guo et al., 2020) | 94.24% | ↓ 0.98 | 266M | 79.6% | 77.1% |
| KPGP (Zhang et al., 2022) | 94.35% | ↓ 0.87 | 273M | 79.0% | 76.9% |
| VQ k1(16,8)/k3(16,2) | 93.17% | ↓ 0.98 | **262.99M** | **79.8%** | **97.4%** |

**VGG16** In the case of CIFAR-10, we adopted a variation of the VGG16 architecture as detailed by (Li et al., 2016). Given that this architecture predominantly utilizes 2D convolutions with $3 \times 3$ kernel sizes, we excluded linear layers from the vector quantization process due to their minuscule size. It should be noted that for VGG16, a single codebook with only 16 entries was found to be inadequate. To further investigate this limitation, we conducted an analysis of codebook usage, as depicted in Figure 2, and compared it between VGG16 and ResNet18. The heatmap analysis reveals a significant disparity in codebook entry utilization between the two architectures. While ResNet18 exhibits a uniform pattern of codebook entry usage, VGG16 appears to struggle in efficiently utilizing many codebook entries, particularly in its later layers. This observation strongly suggests that the distribution of codebook entries required for different layers of VGG16 is not consistent, which could be attributed to the absence of skip connections in the VGG architecture.

To address these suboptimal results, two potential solutions were explored: increasing the number of entries in the codebook or employing multiple codebooks tailored for different depths. Our experimentation involved training two sets of vector-quantized models: 1) A single, larger codebook shared across the entire model. 2) Four separate, smaller codebooks, each dedicated to one of the convolutional blocks. Notably, the last two blocks, both containing the same number of filters, share a common codebook. The results, presented in Table 4, reveal an intriguing trade-off. A model equipped with a single codebook containing 64 entries exhibits marginally improved accuracy compared to a model featuring four codebooks, each with 16 entries. However, it is worth highlighting that the latter option offers greater computational efficiency and slightly decreases memory footprint by reducing the size of the multiplication lookup tables.

A comprehensive evaluation of vector quantized VGG16 on CIFAR-10, considering various codebook sizes, is detailed in Appendix A.5. It is observed that employing a single large codebook, despite yielding slightly better accuracy, is not efficient in terms of FLOPs. The impact of increasing the number of entries ($L$) for the K3 codebook on accuracy is more pronounced in VGG16 compared to ResNet50, primarily due to VGG16's predominant use of $3 \times 3$ convolutions.

Table 4: Effect of having a single large codebook versus multiple smaller codebook for different layers on VGG16 (CIFAR-10)

| K3 codebook (M, B) | Acc. (%) | △ Acc. (%) | FLOPs | FLOPs ↓ (%) | Params | Params ↓ (%) |
|---|---|---|---|---|---|---|
| Original VGG16 | 92.54% | - | 314.43M | - | 14.73M | - |
| (64,1) | 92.08% | ↓ 0.46 | 142.39M | 54.72% | 0.42M | 97.12% |
| 4×(16,1) | 90.86% | ↓ 1.68 | **63.65M** | **79.76%** | **0.42M** | **97.12%** |
| (128,1) | **92.17%** | ↓ 0.37 | 247.38M | 21.33% | 0.43M | 97.11% |
| 4×(32,1) | 91.95% | ↓ 0.59 | 89.89M | 71.41% | 0.43M | 97.11% |

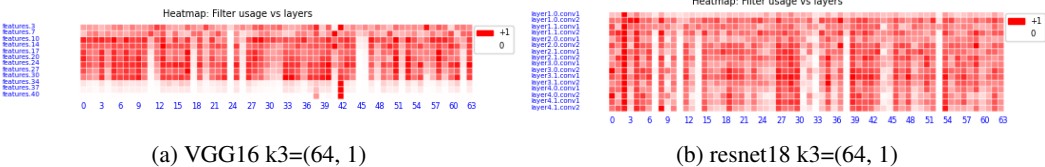

(a) VGG16 k3=(64, 1)  (b) resnet18 k3=(64, 1)

Figure 2: Distribution of codebook usage across different layers (CIFAR-10)

**Comparison with the state of the art:** Table 5 presents a comparative analysis of VQ-VGG16 alongside state-of-the-art (SOTA) methods. The models are grouped according to their computational complexity, as measured by FLOPs. Similar to the performance achieved with ResNet models, our compression ratio outperforms all prior methods considerably. Furthermore, the reduction in accuracy ranks favorably compared to all other approaches, with the exception of DSP (Park et al., 2023).

Table 5: Comparisons of different methods for compressing/accelerating VGG-16 on CIFAR-10 with different pruning ratios. BA represent baseline accuracy.

| ID | BA Acc. (%) | Acc. (%) | △ Acc. (%) | FLOPs ↓ (%) | Params ↓ (%) |
|---|---|---|---|---|---|
| PFEC (Li et al., 2016) | **94.27%** | 93.22% | ↓ 1.07 | 43.3% | 40.7% |
| FPGM (He et al., 2019) | **94.27%** | 92.95% | ↓ 1.32 | 50.4% | 49.9% |
| KPGP (Zhang et al., 2022) | **94.27%** | 93.66% | ↓ 0.61 | 50.4% | 49.9% |
| HRank (Lin et al., 2020a) | 93.96% | 93.43% | ↓ 0.53 | 53.5% | 82.9% |
| VQ (64,1)/(64,1)/(64,1)/(64,1) | 92.54% | 92.20% | ↓ 0.34 | 54.7% | **97.1%** |
| PFEC (Li et al., 2016) | **94.27%** | 92.03% | ↓ 2.24 | 74.0% | 70.9% |
| FPGM (He et al., 2019) | **94.27%** | 88.22% | ↓ 6.05 | 74.4% | 74.9% |
| KPGP (Zhang et al., 2022) | **94.27%** | 92.36% | ↓ 1.91 | 74.4% | 74.9% |
| HRank (Lin et al., 2020a) | 93.96% | 91.23% | ↓ 2.73 | 76.5% | 92.0% |
| DSP (g=2) (Park et al., 2023) | 93.88% | 93.88% | 0.00 | 75.5% | 74.5% |
| DSP (g=4) (Park et al., 2023) | 93.88% | **93.91%** | ↑ **0.03** | **77.8%** | 76.6% |
| VQ (32,1)/(16,1)/(16,1)/(16,1) | 92.54% | 91.86% | ↓ 0.68 | 76.0% | **97.1%** |

## 4.2 IMAGENET

**Comparison with SOTA methods**: We trained two vector-quantized ResNet50 models ($M = 16, M = 32$) on the ImageNet dataset each employing three codebooks tailored for $1 \times 1$ convolution $(M, 8)$, $3 \times 3$ convolution $(M, 1, 3, 3)$, and the linear classification head $(M, 8)$. The number of entries in the codebook, represented by $M$, is indicated in parentheses in Table 6. In the case of vector quantized Vision Transformer (ViT) models, two codebooks of dimensions $(M, 8)$ were utilized. These codebooks were assigned to the in-projection matrices of the multihead attention layers and the MLP layers, respectively. The vector quantized ResNet50 model demonstrates a substantial performance advantage over all existing state-of-the-art methods, excelling in terms of both computational complexity (measured in FLOPs) and model parameters. Similarly, the vector quantized ViT model exhibits remarkable efficiency and compactness, surpassing all other state-of-the-art methods in these aspects as well. Our observations reveal that initializing from pretrained model weights yields superior results. The results enclosed within parentheses for our models denote those obtained through the utilization of pretrained model weights.

Table 6: Comparisons of different methods for compressing/accelerating ResNet50/ViT on ImageNet with different pruning ratios. BA represent baseline accuracy. Numbers is parenthesis present models initialized from a pretrained model.

| ID | BA Acc. (%) | Acc. (%) | △ Acc. (%) | FLOPs ↓ (%) | Params ↓ (%) |
|---|---|---|---|---|---|
| FPGM (He et al., 2019) | 76.15% | 74.83% | ↓ 1.32 | 53.5% | - |
| Rethink (Liu et al., 2018) | 76.13% | 73.90% | ↓ 2.23 | 50.0% | - |
| PScratch (Wang et al., 2020b) | **77.20%** | 75.60% | ↓ 1.60 | 51.2% | 63.9% |
| SG-CNN (Guo et al., 2020) | 76.13% | 75.20% | ↓ 0.93 | 53.3% | 53.5% |
| GCP (Zhao & Luk, 2019) | 76.15% | 74.10% | ↓ 2.05 | 54.1% | 45.9% |
| KPGP (Zhang et al., 2022) | 76.15% | 75.58% | ↓ 0.57 | 44.3% | 44.0% |
| SCOP (Tang et al., 2020) | 76.15% | 75.26% | ↓ 0.89 | 54.6% | 51.8% |
| VQ ResNet50 (16) | 76.82% | 70.40% (74.61%) | ↓ 6.42 (2.21) | **74.8%** | **96.8%** |
| VQ ResNet50 (32) | 76.82% | **71.85% (76.35%)** | **↓ 4.97 (0.54)** | 64.7% | **96.8%** |
| VTP (Zhu et al., 2021) | **81.8%** | 80.7% | ↓ 1.1 | 43.2% | 45.2% |
| PatchSlimming (Tang et al., 2022) | **81.8%** | 81.6% | ↓ 0.2 | 46.6% | - |
| WDPruning (Yu et al., 2022) | **81.8%** | 80.76% | ↓ 1.04 | 43.4% | 35.0% |
| S$^2$ViTE (Chen et al., 2021) | **81.8%** | **82.22%** | **↑ 0.42** | 33.13% | 34.41% |
| VQ ViT-B (16) | 78.51% | 75.30% (76.71%) | ↓ 3.21 (1.79) | **84.65%** | **97.79%** |

### 4.3 GAINING FURTHER EXPERIMENTAL INSIGHTS

This section is dedicated to a series of experiments designed to delve into the attributes and capabilities of vector quantized deep neural networks (VQ-DNNs). In our initial experiment, details of which are provided in the Appendix A.2, we conducted an examination of the significance of individual entries within the codebook, aiming to determine whether each entry holds equal importance. In the subsequent experiment, we sought to gain insights into how VQ networks employ their codebooks across various layers. Additional information can be found in the Appendix A.3. Another experiment we conducted investigating the transferability of learned codebooks between distinct models and datasets, as elaborated upon in the Appendix A.6. Vector quantized deep neural networks can be seamlessly integrated with various other methods for compressing and accelerating deep neural networks. One prominent category among these methods is low-bit quantization techniques. To validate the compatibility of our approach with low-bit precision methods, we applied 8-bit quantization to several of our trained models. Detailed results are available in the Appendix A.4. Lastly, we conducted an analysis to discern the features learned by VQ-DNNs and compared them with those acquired by standard networks in Appendix A.7.

## 5 CONCLUSION

This research introduces Vector Quantized Deep Neural Networks (VQ-DNNs), which excel in achieving state-of-the-art compression and acceleration rates. Unlike traditional quantization methods that reduce parameter precision, VQ-DNNs take a unique approach by learning a codebook during training. This codebook consists of segments that exclusively form the network's weights, effectively replacing model parameters with indices referencing the codebook. Frequent repetition of codebook values at each layer allows for an efficient VQ-DNN implementation where each layer's input is multiplied (or convolved) with the entire codebook, reducing subsequent operations into lookup procedures. As a result, both the parameter count and Floating-Point Operations (FLOPs) are reduced, surpassing current compression techniques significantly. Moreover, VQ-DNNs achieve competitive FLOP efficiency compared to leading methods. Additionally, VQ-DNNs are compatible with various other methods, such as lower-precision operations, enabling their integration to enhance efficiency. This new paradigm introduces a promising direction in the existing literature, potentially augmenting current approaches to deep learning model compression and acceleration.

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

# A  APPENDIX

## A.1  TRAINING RECIPE

In our experimentation with the CIFAR-10 dataset, we implemented models employing the stop gradient operation and integrated three distinct loss terms, as comprehensively detailed in Section 3.3. For the ImageNet dataset, we employed an exponential moving average estimation approach, as described in Section 3.3, primarily due to its faster convergence. However, when subjecting this method to CIFAR-10, we observed no substantial variations in terms of accuracy in comparison to the prior approach.

Notably, we discerned suboptimal performance when employing the SGD optimizer with the vector quantized model. Conversely, our experimental results demonstrated that the AdamW optimizer (Loshchilov & Hutter, 2017) exhibits a satisfactory performance. Consequently, we have adopted the AdamW optimizer as the preferred choice for all our experiments, regardless of whether we are training a vector quantized model or a baseline model. Moreover, we incorporated RandAugment as our chosen data augmentation technique, as detailed in (Cubuk et al., 2020). Our training process included the implementation of a one-cycle learning rate schedule following the approach outlined in (Smith & Topin, 2019), with PyTorch's (Paszke et al., 2017) default settings. The remaining hyperparameters employed in our experiments are provided in Table A.1 for reference.

Table A.1: Hyperparameters used in our experiments

| Dataset | Architecture | Epochs | Batch size | Max learning rate | Weight decay |
|---|---|---|---|---|---|
| CIFAR-10 | ResNet50/18, VGG16 | 200 | 128 | 0.005 | 0.0005 |
| ImageNet | ResNet50 | 90 | 256 | 0.003 | 0.0001 |
| ImageNet | ViT_b_16 | 90 | 1024 | 0.003 | 0.0001 |
| EuroSat | ResNet50 | 90 | 256 | 0.003 | 0.0001 |

## A.2 EFFECT OF REMOVING FILTERS

In this experimental investigation, we systematically execute a codebook removal procedure and subsequently assess its influence on the model's accuracy. For every weight index initially associated with the target codeword, we conduct a search within the remaining codebook entries to identify and replace it with the nearest available codeword. The outcomes, depicted graphically in Figure A.1, yield notable insights.

Notably, when employing a codebook with a mere 16 entries, the elimination of each codebook entry results in a substantial deterioration in accuracy. However, with the utilization of a 64-entry codebook, the decline in accuracy is significantly less pronounced. This phenomenon hints at the existence of redundancy within the codebooks, with many codewords possessing close counterparts.

Of particular interest is the absence of a discernible correlation between the frequency of codebook usage and its subsequent impact on accuracy. This observation holds true for the K1 codebook as well. Due to limitations in presentation space, we exclusively present the findings related to the K3 codebook in this context.

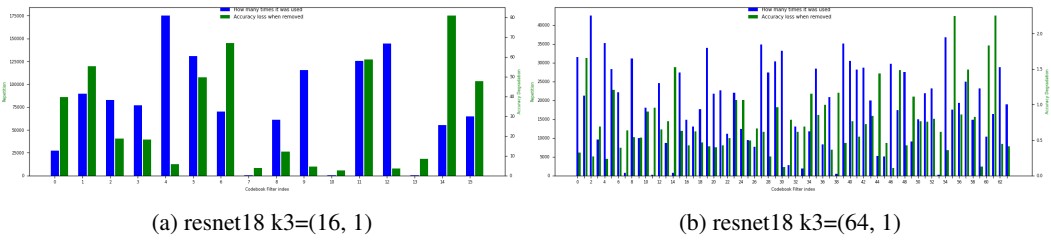

(a) resnet18 k3=(16, 1)           (b) resnet18 k3=(64, 1)

Figure A.1: The effect of removing each codebook on the accuracy of ResNet18 trained on CIFAR-10

## A.3 DISTRIBUTION OF FILTER USAGE

To investigate the utilization pattern of codebooks across various layers in VQ-DNNs, we present a visualization illustrating the distribution of unique filters applied at each layer, as presented in Figure A.2. Notably, a substantial proportion of codebook entries are employed consistently across all layers. This observation stands in contrast to conventional deep neural networks (DNNs), where prior research has underscored distinctions between filters in initial layers and those in deeper layers (Gavrikov & Keuper, 2022).

The implication of this finding is indicative of the model's ability, particularly in cases where skip connections are incorporated, to exhibit a high degree of adaptability. It efficiently accomplishes tasks by recurrently utilizing a relatively compact subset of filters.

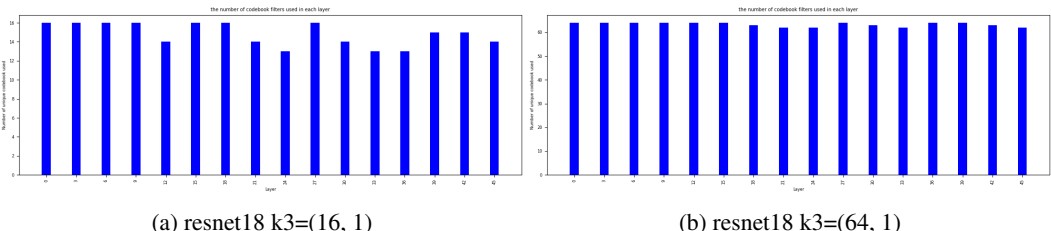

(a) resnet18 k3=(16, 1)           (b) resnet18 k3=(64, 1)

Figure A.2: How many unique codebooks are used at each layer for a ResNet18 trained on CIFAR-10

## A.4 8-BIT QUANTIZATION

Table A.2 presents the outcomes obtained from the application of 8-bit quantization to a vector quantized model. This quantization process encompasses both weight parameters (e.g., codebook entries)

and intermediate activation values. Across the majority of instances, the reduction in accuracy from the initial vector quantized model remains less than one percent. This observation underscores the potential for leveraging vector quantization in conjunction with low-bit quantization as a notably effective strategy to enhance computational speed and reduce memory usage.

Table A.2: 8-bit quantization of vector quantized models trained on CIFAR-10

| ID | Codebook | △ 8-bit Acc. (%) |
|---|---|---|
| ResNet18 | - | ↓ 0.56% |
| VQ_ResNet18 | k1(16,8)/k3(16,8) | ↓ 1.54% |
| VQ_ResNet18 | k1(16,8)/k3(16,4) | ↓ 0.80% |
| VQ_ResNet18 | k1(16,8)/k3(16,2) | ↓ **0.07%** |
| VQ_ResNet18 | k1(16,8)/k3(16,1) | ↓ 0.39% |
| VQ_ResNet18 | k1(32,8)/k3(32,8) | ↓ 1.17% |
| VQ_ResNet18 | k1(32,8)/k3(32,4) | ↓ 0.67% |
| VQ_ResNet18 | k1(32,8)/k3(32,2) | ↓ 0.55% |
| VQ_ResNet18 | k1(32,8)/k3(32,1) | ↓ 0.90% |
| VGG16 | - | ↓ 0.72% |
| VQ_VGG16 | k3(64,1) | ↓ **0.39%** |
| VQ_VGG16 | k3(16,1)/(16,1)/(16,1)/(16,1) | ↓ 0.47% |
| VQ_VGG16 | k3(128,1) | ↓ 0.77% |
| VQ_VGG16 | k3(32,1)/(32,1)/(32,1)/(32,1) | ↓ 0.47% |

## A.5 VGG16 ON CIFAR-10

A thorough assessment of vector quantized VGG16 model performance on the CIFAR-10 dataset, considering diverse codebook sizes, is presented in Table A.3. Notably, the adoption of a single, extensive codebook, despite yielding a marginally superior accuracy, is found to be inefficient in terms of Floating-Point Operations (FLOPs).

Furthermore, it is observed that the impact of increasing the codebook size parameter ($M$) for the K3 codebook is more pronounced in the VGG16 model as compared to the ResNet50 model. This difference is primarily attributed to the architectural distinction between the two models, with VGG16 predominantly employing $3 \times 3$ convolutions.

Table A.3: Effect of having a single large codebook versus multiple smaller codebook for different layers on VGG16 (CIFAR-10)

| K3 codebook (M, B) | Acc. (%) | △ Acc. (%) | FLOPs | FLOPs ↓ (%) | Params | Params ↓ (%) |
|---|---|---|---|---|---|---|
| Baseline | 92.54% | - | 314.43M | - | 14.73M | - |
| (64,1)/(64,1)/(64,1)/(64,1) | **92.20%** | ↓ **0.34** | 142.39M | 54.72% | 0.43M | 97.11% |
| (48,2)/(48,2)/(48,2)/(48,2) | 91.06% | ↓ 1.48 | 98.84M | 68.57% | 0.22M | 98.49% |
| (48,1)/(48,1)/(48,1)/(48,1) | 91.81% | ↓ 0.73 | 116.14M | 63.06% | 0.43M | 97.11% |
| (32,8)/(32,8)/(32,8)/(32,8) | 84.74% | ↓ 7.80 | 59.61M | 81.04% | 0.08M | 99.49% |
| (32,4)/(32,4)/(32,4)/(32,4) | 89.10% | ↓ 3.44 | 63.94M | 79.66% | 0.12M | 99.17% |
| (32,2)/(32,2)/(32,2)/(32,2) | 91.21% | ↓ 1.33 | 72.59M | 76.91% | 0.22M | 98.49% |
| (32,1)/(32,1)/(32,1)/(32,1) | 91.95% | ↓ 0.59 | 89.89M | 71.41% | 0.43M | 97.11% |
| (16,8)/(16,8)/(16,8)/(16,8) | 82.20% | ↓ 10.34 | **33.37M** | **89.39%** | 0.07M | 99.52% |
| (16,4)/(16,4)/(16,4)/(16,4) | 87.64% | ↓ 4.90 | 37.69M | 88.01% | 0.12M | 99.19% |
| (16,2)/(16,2)/(16,2)/(16,2) | 89.25% | ↓ 3.29 | 46.34M | 85.26% | 0.22M | 98.50% |
| (16,1)/(16,1)/(16,1)/(16,1) | 90.86% | ↓ 1.68 | 63.65M | 79.76% | 0.42M | 97.12% |
| (32,1)/(16,1)/(16,1)/(16,1) | 91.86% | ↓ 0.68 | 75.44M | 76.01% | 0.42M | 97.12% |
| (16,1)/(32,1)/(16,1)/(16,1) | 91.24% | ↓ 1.30 | 69.54M | 77.88% | 0.42M | 97.12% |
| (16,1)/(16,1)/(32,1)/(16,1) | 91.23% | ↓ 1.31 | 68.95M | 78.07% | 0.42M | 97.12% |
| (16,1)/(16,1)/(16,1)/(32,1) | 91.34% | ↓ 1.20 | 66.89M | 78.73% | 0.42M | 97.12% |
| (128,8) | 88.50% | ↓ 4.04 | 217.10M | 30.96% | 0.08M | 99.49% |
| (128,4) | 90.26% | ↓ 2.28 | 221.42M | 29.58% | 0.12M | 99.17% |
| (128,2) | 91.60% | ↓ 0.94 | 230.07M | 26.83% | 0.22M | 98.49% |
| (128,1) | 92.17% | ↓ 0.37 | 247.38M | 21.33% | 0.43M | 97.11% |
| (96,8) | 88.07% | ↓ 4.47 | 164.60M | 47.65% | 0.07M | 99.50% |
| (96,4) | 89.83% | ↓ 2.71 | 168.93M | 46.28% | 0.12M | 99.18% |
| (96,2) | 91.15% | ↓ 1.39 | 177.58M | 43.52% | 0.22M | 98.50% |
| (96,1) | 91.59% | ↓ 0.95 | 194.88M | 38.02% | 0.42M | 97.12% |
| (64,8) | 87.08% | ↓ 5.46 | 112.11M | 64.35% | 0.07M | 99.52% |
| (64,4) | 89.47% | ↓ 3.07 | 116.43M | 62.97% | 0.12M | 99.19% |
| (64,2) | 90.95% | ↓ 1.59 | 125.09M | 60.22% | 0.22M | 98.50% |
| (64,1) | 92.08% | ↓ 0.46 | 142.39M | 54.72% | 0.42M | 97.12% |
| (16,8) | 81.41% | ↓ 11.13 | 33.37M | 89.39% | **0.07M** | **99.54%** |
| (16,4) | 86.69% | ↓ 5.85 | 37.69M | 88.01% | 0.12M | 99.20% |
| (16,2) | 89.47% | ↓ 3.07 | 46.34M | 85.26% | 0.22M | 98.51% |
| (16,1) | 91.09% | ↓ 1.45 | 63.65M | 79.76% | 0.42M | 97.12% |

## A.6 ARE CODEBOOKS TRANSFERABLE?

In this section, our objective is to investigate the transferability of codebooks initially trained for one task to another task. Our approach involved initially training a ResNet50 model on the ImageNet dataset. Subsequently, we transposed the codebook entries, while excluding the model parameters and indexes, into a new model. We maintained the original codebook's entries by fixing them during training on the EuroSat dataset, a deliberate choice owing to its notably distinct image distribution in comparison to ImageNet.

It is essential to highlight that the effectiveness of this approach hinged on codebook rescaling. Due to the significantly larger range of the trained codebook values in contrast to randomly initialized model weights, the training process exhibited convergence issues and considerable instability. Consequently, we undertook a rescaling of the codebook weights to confine them within the range of -1 to 1.

Notably, a non-vector-quantized (non-VQ) ResNet50 model, when directly trained on the EuroSat dataset, achieves an accuracy of $96.71\%$. Remarkably, a model incorporating the fixed, transferred filters from an ImageNet task attains an accuracy of $95.85\%$, demonstrating a negligible difference of less than one percentage point. This compelling result strongly implies the generalizability of trained codebooks across distinct tasks.

## A.7 WHAT FEATURES DO VECTOR QUANTIZED MODELS LEARN?

A potential concern arises from the deployment of a single codebook throughout the entirety of ResNet models, encompassing those incorporating skip connections. This concern is related to the possibility that the vector quantized model may predominantly acquire and propagate localized features and patterns across the model via these skip connections. To gain deeper insights into the nature of features acquired by the vector quantized ResNet50, we applied GradCAM++ (Chattopadhay et al., 2018) to generate saliency maps for input images. This analysis was conducted utilizing the implementation provided by (Gildenblat & contributors, 2021).

As illustrated in Figure A.3, the saliency maps derived from various samples representing different classes exhibit noteworthy similarities when comparing ResNet50 (second row) with vector quantized ResNet50 (third row). Intriguingly, the VQ ResNet50 demonstrates a slightly heightened emphasis on the target object in comparison to ResNet50. This observation distinctly underscores that, even with a relatively compact codebook, VQ models exhibit learning capabilities on par with those of conventional models.

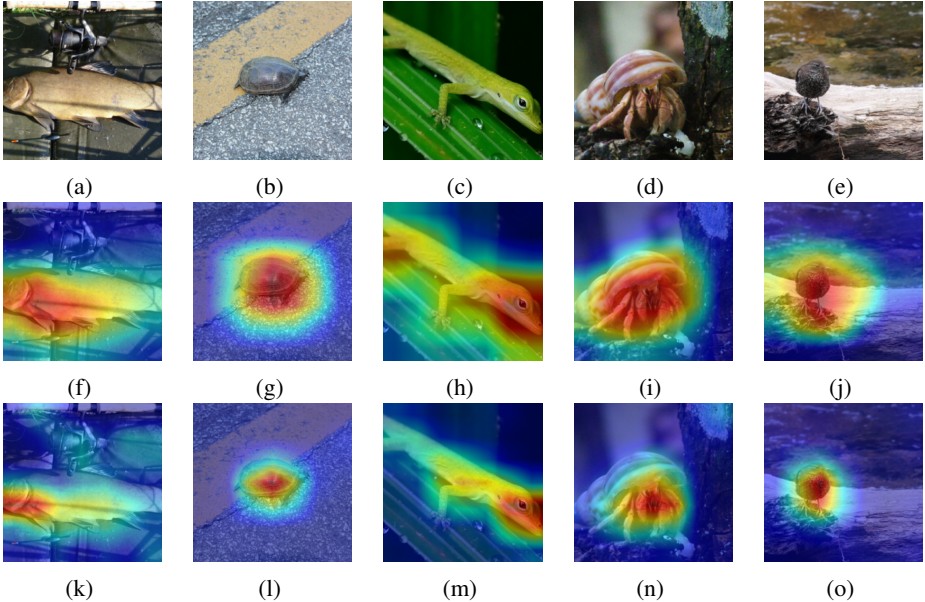

Figure A.3: GradCam++ on five different images. The first row is the original image. The second and third rows are the results of ResNet50 and VQ ResNet50, respectively. Although both ResNet50 and VQ-ResNet50 capture almost entire area of the objects, VQ-ResNet50 is slightly more localized.

## A.8 COMPARISON WITH DKM

In this section, we conduct a comparative analysis between our results and those presented in the DKM study (Minsik et al., 2022). Throughout the main body of the paper, to be consistent with standard practice in the literature, we report model size in terms of parameter count. We assume a standard parameter size of 32 bits, with each index accounting for 1 Byte, equivalent to one-fourth of a parameter. However, it's worth noting that in (Minsik et al., 2022), model size is reported in megabytes (MB). In this section, we adhere to this convention and report results in MB for fair comparison. Furthermore, we compute the exact number of bits required for each index, which consistently falls below 1 Byte. As illustrated in Table A.4, our methodology demonstrates superior performance in terms of network size compared to DKM. This discrepancy primarily arises from our utilization of a single codebook across the entire model, whereas DKM employs separate codebooks for each layers. It is important to mention that the DKM approach starts its training process using a pre-trained model. Consequently, the results we present in this context are based on training that initiated from a pre-trained weights and the baseline accuracy is the accuracy of the pre-trained model. To have a fair comparison, we employed an identical set of pretrained weights as utilized in the DKM method. This particular set of weights yielded an accuracy rate of 76.1%. It is noteworthy that in the primary content of the paper, we had utilized an alternative version of pretrained model weights, which achieved an accuracy rate of 80.86%. Additionally, it should be acknowledged that we have achieved marginally improved results while employing a training duration of 90 epochs, as opposed to the 200 epochs utilized in the DKM method.

Table A.4: Comparison with DKM (Minsik et al., 2022)

| ID | BA Acc. (%) | Acc. (%) | △ Acc. (%) | FLOPs ↓ (%) | Params (MB) | Params ↓ (%) |
|---|---|---|---|---|---|---|
| DKM (cv:6/6, fc:6/4) | **76.10%** | 74.50% | ↓ 1.60 | - | 3.32 | 96.59% |
| VQ ResNet50 (M:32, B:8) | **76.10%** | **75.31%** | ↓ **0.79** | **64.7%** | **0.92** | **99.05%** |

