# OpenReview forum: "COMPRESSION AND ACCELERATION OF DEEP NEURAL NETWORKS: A VECTOR QUANTIZATION APPROACH"
_ICLR.cc/2024/Conference — Submitted to ICLR 2024_

### Official Review · Reviewer_hXiT · 2023-10-29

**Soundness:** 2 fair
**Presentation:** 2 fair
**Contribution:** 2 fair
**Rating:** 3
**Confidence:** 5

**Summary:**

The paper introduces a vector quantization technique that involves partitioning the weight matrix into several segments and constructing a global codebook. This approach quantizes the weights into many segmented vectors based on the global codebook. The authors claim that this method can significantly reduce the storage requirements of the model, as the weights only need few bits to store the codebook indices. Additionally, it can substantially decrease the computational operations of the model.

**Strengths:**

Applying the global codebook and segmented vector quantization can significantly compress the model size.

**Weaknesses:**

The content of this paper is not sufficiently clear and complete. Many aspects are either missing or are ambiguously addressed. For instance, the paper lacks details on how to handle activation values, how to quantize weights into segmented vectors, how to update quantized weights and codebooks, and how to perform matrix multiplications and convolutions on quantized weights and activation values. These critical aspects are left unaddressed.
In terms of the paper's novelty and motivation, the primary source of confusion lies in the paper's failure to clarify what problem in quantization or vector quantization it aims to address or improve. This paper claims to apply a global codebook to the entire model, as opposed to layer-wise application as did in previous studies, which theoretically could improve compression rates but might significantly impact model accuracy. However, the authors do not discuss this issue in the paper. Lastly, regarding the compression of operation counts, the method utilizing codebooks and indices primarily offers theoretical compression, and practical acceleration is challenging to achieve. This issue should be argued in the paper.

**Questions:**

1、How are activation values quantized?

2. The paper employs a global codebook for the entire model instead of using one per layer. Does this affect accuracy?

3. In the experimental results, the reported FLOPS and Params values are purely theoretical, correct? Can you discuss the practical acceleration or its applicability? Additionally, how was the base accuracy determined? Is the primary metric accuracy or accuracy reduction? Why is accuracy reduction considered more effective?

---

> ### Author Response · Authors · 2023-11-13
>
> Thank you for providing valuable feedback on our paper. We appreciate your thoughtful comments. It appears that some of the confusion may arise from the terminology used in our method, namely "vector quantization," which, despite containing the term "quantization," does not align with traditional low-precision encoding techniques commonly associated with quantization. we chose to adhere to the established concept of "vector quantization" in the data compression and deep learning literature and we didn't coin a new term.
>
> As explicitly stated in our paper, our approach involves the vector quantization of weight parameters exclusively. We want to emphasize that we do not extend this quantization to activations. Our study demonstrates that employing a compact codebook for the entire model yields acceptable accuracy, and any reduction in accuracy is not deemed significant. Importantly, this reduction in accuracy is accompanied by a considerable decrease in both floating-point operations (FLOPs) and memory usage, as substantiated in our theoretical analysis.
>
> Regarding the practical speed-up, we would like to draw your attention to our response to question 1 from reviewer 1, where we delve into the specifics of achieving practical acceleration with our proposed method. We acknowledge the importance of discussing the practical implications.
>
> We genuinely value your input and believe that these clarifications will contribute to a more coherent and comprehensive presentation of our work. If our paper is accepted, we are committed to incorporating these refinements to enhance the overall clarity and understanding of our method.
>
> Here are the responses to the questions:
>
> 1: In the context of this paper, it's important to note that activation values remain in their full precision without quantization. The quantization specifically applies to the weight parameters of the model, which undergo vector quantization. This means that these weights are constrained to discrete values defined by the codebook. To enhance clarity, if our paper is accepted, we intend to provide a more detailed clarification regarding the differentiation between weight parameters and activations in terms of quantization.
>
> 2: In our experiments, detailed in Appendix A.5 using the VGG model, we investigated the impact of employing either four codebooks or a single global codebook. Interestingly, we observed only a marginal difference in performance. Given this, we chose to emphasize the more extreme scenario of employing a single global codebook, which achieves the highest compression rate. We plan to provide additional clarification on this choice in the main body of the paper, should it be accepted.
>
> 3: In addressing the question regarding FLOPs and Params values in our experimental results, it's crucial to note that these values, as well as those in the cited papers, are primarily theoretical. Achieving real acceleration demands either hardware implementation or alternative methods such as implementing CUDA modules. We provide a more comprehensive response to this concern in our reply to the first reviewer's initial questions. This discussion will be clarified in the final paper if accepted.
>
> The determination of base accuracy varies depending on the method employed. For post-training approaches applied to pre-trained models, the base accuracy represents the model's accuracy before pruning. As we lack control over external methods, we report the specifics of their procedures. For instance, the base ResNet50/ImageNet model in PScratch (Wang et al., 2020b) exhibited an accuracy of 77.20%, while SG-CNN (Guo et al., 2020) had 76.13%. In the case of methods like our VQ method, which involves training a model from scratch, we ensure consistency by training two models—one non-VQ as a baseline and one VQ model—using identical hyperparameters (epochs, batch size, data augmentations, etc.).
>
> The emphasis on accuracy reduction over absolute accuracy stems from the inherent variability in base accuracies across methods. Given constraints such as training ResNet50/Imagenet for 90 epochs with a batch size of 256 (refer to Appendix A.1 for details), our baseline accuracy is naturally lower than approaches utilizing well-trained public models. Consequently, we believe assessing the degree of accuracy degradation provides a fairer comparison, considering the varied starting points of different methods, rather than fixating on absolute accuracy figures.

---

### Official Review · Reviewer_JeiW · 2023-11-01

**Soundness:** 3 good
**Presentation:** 2 fair
**Contribution:** 2 fair
**Rating:** 5
**Confidence:** 3

**Summary:**

This paper suggests using vector quantization for compressing the weights of convolutional layers in Convolutional Neural Networks (CNNs) and attention/Multilayer Perceptron (MLP) layers in Vision Transformers (ViTs). The weights are associated with learned codebooks. During the inference stage, computations occur between the features and the codebooks, which can significantly reduce parameters and Floating Point Operations Per Second (FLOPs). Experiments provide a comparison with the state-of-the-art.

**Strengths:**

This work utilizes vector quantization to compress DNNs, it is good to consider both CNNs and ViTs in the experiments.

**Weaknesses:**

1. The figures in this paper are really vague. In Figure 1, it is better to mark the size of the weight matrices and demonstrate which layers in which model are utilized to do the visualization.
2. Experimental results are listed without any highlight, which is very unclear.
3. The idea of adopting VQ in compressing DNNs is not new, so more effort could be made including giving the guidelines for designing the hyperparameters such as the group numbers.

**Questions:**

1. It appears that the authors directly use the computation results from another VQ paper to determine the reduction in FLOPs. Is this method accurate for DNN compression? A related question is, given that the input and intermediate features in DNNs vary and are not predetermined, how are the lookup tables obtained in this context?
2. Besides #parameters and FLOPs, how about the inference time? Could VQ accelerate DNN inference?

---

> ### Author Response · Authors · 2023-11-13
>
> Here are our brief responses to the reviewer's comments:
>
> Weaknesses
>
>
> 1:
> Thank you for highlighting this issue. In our revised draft, we have made significant enhancements to Figure 1 to provide a more lucid representation of the Vector Quantization concept. We've taken care to align the figure closely with the notations employed in the paper, ensuring a clearer depiction of the size of matrices. These adjustments aim to eliminate any ambiguity and enhance the overall comprehensibility of the figure.
>
> 2:
> Thank you for bringing this to our attention. We have addressed the clarity concern by incorporating highlights in the tables in our revised draft. This adjustment aims to improve the visual distinction of experimental results, making it easier for readers to interpret and understand the presented data.
>
> 3:
> Our sincere thanks for your comment. While it's acknowledged that clustering weights for DNN compression isn't a novel idea, our paper uniquely contributes by being the first, to the best of our knowledge, to directly apply Vector Quantization (VQ) to neural network weights. This distinction is crucial as it pioneers a new application of VQ in the context of weight compression. Our approach introduces two key contributions: firstly, demonstrating the effectiveness of a single compact codebook in encoding layers of large architectures, exemplified by ResNet-50; and secondly, emphasizing the efficiency gains in computation through the use of a small, single codebook. This innovation allows for streamlined operations, multiplying the entire codebook by inputs in each layer, thereby circumventing redundant calculations. We acknowledge the existing work in weight clustering and stress that our unique application of VQ to network weights sets our approach apart, offering a fresh perspective and valuable contributions to the field.
>
> Questions:
>
>
> 1:
> Thank you for your thoughtful inquiry. While our FLOP computation isn't directly borrowed from another paper, we have adhered to established practices in the literature for consistency. In Section 3.4, we provide a detailed explanation of our FLOP calculation methodology.
>
> Concerning lookup tables, it's important to note that these are determined on a per-layer basis. Unlike the global nature of the codebook, which remains consistent across layers, the lookup matrix is computed uniquely for each layer. This approach is necessary due to the varying nature of input and intermediate features in different layers of DNNs.
>
> During inference, at each layer, the input undergoes multiplication with the codebook, resulting in the creation of the lookup table. Once the layer's output is generated, the corresponding lookup table becomes obsolete and is discarded. This process is repeated at every layer. In the event of acceptance, we plan to provide further clarification on this process in the paper to ensure a comprehensive understanding of the lookup table generation.
>
> 2:
> Thank you for your inquiry regarding the inference time. If by "inference time" you are referring to the "inference phase," then our response is as follows: The FLOPs computation in the paper specifically pertains to the inference time, demonstrating that Vector Quantization (VQ) does indeed accelerate Deep Neural Networks (DNNs) during this phase.
>
> It's crucial to note a distinction between training and inference. During training, the original weight matrix remains in place, and the forward pass temporarily replaces weight parameters with the closest ones in the codebook. This process is dynamic during training, leading to a slightly slower training speed compared to inference. However, this is not a concern for the inference time, where the model weights are fixed, and a pre-computed index to the closest codebook is applied only once.
>
> While the FLOPs presented in the paper are specifically for the inference time, it's important to acknowledge that training involves a dynamic mapping process, which can't be pre-computed. No prior literature has reported the computation overhead during training or post-training for their approaches, making a direct comparison challenging. If the paper is accepted, we are committed to providing additional clarification on this process.
>
> If by "inference time," you are referring to the actual time spent on inference, please refer to our response to the first question from reviewer 1 for more details.

---

### Official Review · Reviewer_Be5z · 2023-11-08

**Soundness:** 3 good
**Presentation:** 3 good
**Contribution:** 3 good
**Rating:** 6
**Confidence:** 4

**Summary:**

This work proposed a novel idea of applying vector quantization to the weights of deep neural networks. Vector quantization means a weight tensor in neural networks can be represented as the combination of multiple vectors, and the pattern types of the vectors are finite. Similar to network pruning and network quantization, Conducting vector quantization for a neural network is favorable for saving memory occupancy and potentially accelerating inference.

**Strengths:**

1. The idea of conducting vector quantization on weights of neural networks is of novelty.
2. The proposed training algorithm to form vector-quantized neural networks is simple yet effective

**Weaknesses:**

1. The conclusion regarding the speed-up performance is not convincing. There is no practical speed-up results presented in the work and the reported FLOPs reduction is only calculated theoretically. Moreover, traditional dense matrix multiplication has been highly optimized in recent years on modern GPUs, e.g., cuDNN. Whether or not a network with vector quantization infers faster than its dense counterparts on modern GPUs remains uncertain. Thus, the lack of inferring time and practical speedup results does not adequately justify their work.

2. Some aspects of the VQ-DNN illustration require further clarification. For instance, the specific process of updating W_jb and e in equation (1) needs to be clarified. Are they updated alternately? Additionally, the method of mapping weights to the codebook is not clear. Does the mapping change during training? It is necessary to provide more detailed explanations for these concerns.

3. The experimental section is confusing probably due to the unclear selection of compared methods. The related work section does not introduce the compared methods in Table 3, 5, and 6, making it difficult to understand the state of the arts. Furthermore, the authors introduce four categories of model compression in the related work section, such as pruning, and quantization, but they do not present the performance comparison between them and the proposed methods.

4. The experimental setup could be improved. ResNet50 is not reasonable for being used on CIFAR10 since it is too large and prone to overfitting, potentially making the experimental results noisy. Additionally, all experiments are focused on classification. The proposed method should be extended to other tasks to provide more generalized results.

5. The quality of figures and tables should be improved. For example, the quality of Figure 2 is poor, which makes it difficult to capture useful information. Similarly, the proposed method's key results in the tables should be highlighted for better readability.

**Questions:**

Please see the weaknesses for revision.

---

> ### Author Response · Authors · 2023-11-13
>
> Here are our brief responses to the reviewer's comments:
>
> 1:
> We appreciate the reviewer's feedback on the lack of practical validation in the paper. We want to clarify that our decision to focus on theoretical aspects is intentional and aligns with the specific scope defined for this study. While we acknowledge the importance of empirical evidence, we deliberately leave the exploration of practical implementation and performance to future research. The theoretical analysis presented here serves as a foundation for potential follow-up studies aimed at empirically validating and optimizing our approach.
>
> We recognize the optimization of traditional dense matrix multiplication on modern GPUs, such as cuDNN. However, we believe that our emphasis on theoretical foundations is a roadmap for future optimizations.
>
> It's important to note that our approach is not unique in this aspect; many SOTA papers in the literature report FLOPs without immediate practical validation (Li et al., 2016; He et al., 2019; Zhang et al., 2022; Lin et al., 2020a; Park et al., 2023; Liu et al., 2018; Wang et al., 2020b; Guo et al., 2020; Zhao & Luk, 2019; Tang et al., 2020; Zhu et al., 2021; Yu et al., 2022; Chen et al., 2021).
>
> Implementing custom Cuda modules may be possible, but a fair comparison would require similar efforts for all previous works, surpassing our current resources. To the best of our knowledge, no previous papers in this domain have implemented Cuda modules or hardware versions of their methods, and this challenge has been acknowledged by reviewers in the field, as exemplified in the discussion regarding "TMI-GKP: Revisit Kernel Pruning with Lottery Regulated Grouped Convolutions" (available at https://github.com/henryzhongsc/lottery_regulated_grouped_kernel_pruning#discussion-regarding-inference-speedup), which was accepted at ICLR 2022 without further implementation.
>
> 2:
> While we commit to providing more detailed explanations in a revised version if the paper is accepted, we offer the following insights for the current discussion:
>
> In eq(1), the update operations for W_jb and e are indeed distinct and are performed simultaneously at the end of each batch. This simultaneous update is facilitated by the use of stop-gradient operators, as described in the original paper by Van Den Oord et al. (2017).
>
> Concerning the mapping of weights to the codebook, it is essential to note that this mapping evolves during training. Specifically, the weight stored in the original model (W_jb) undergoes a comparison with the entire codebook and is subsequently replaced with the nearest codeword. This process essentially treats the original weight as an index. As training progresses, these weights undergo updates, causing the mapping to change dynamically.
>
> 3:
> In our related work section, we aimed for comprehensiveness by covering all categories of model compression found in the literature. Certain methods, such as knowledge distillation or matrix decomposition approaches, have not been included as their popularity is declining due to the superior performance of pruning methods.
>
> Although our method bears the name "vector quantization," it is distinct from the more prevalent category of low-precision parameter methods or traditional quantization. The terminology may introduce some confusion, but we chose to adhere to the established concept of "vector quantization" in the data compression and deep learning literature. Traditional quantization approaches often measure metrics, such as energy consumption (Hubara et al., 2016), or actual latency (Jacob et al., 2018; Lee & Nirjon, 2019), which may not be directly comparable to our method or other pruning methods. Notably, SOTA pruning methods typically compare their performance exclusively with other pruning methods in the literature, given the intrinsic differences in evaluation metrics.
>
> In the event of acceptance, we are committed to refining the related work section to explicitly cover the papers with which we compare our approach, providing a more comprehensive and coherent understanding of the state-of-the-art landscape in model compression.
>
> 4:
> Our choice of experimental configuration aligns with prevalent practices in the literature, where SOTA papers commonly focus on CIFAR and ImageNet classification tasks, using VGG and ResNet for evaluation (Zhang et al., 2022; Park et al., 2023; He et al., 2019; Wang et al., 2020b). While there are exceptions (Lin et al., 2020a; Tang et al., 2020), the majority of ViT pruning studies adhere to the standard practice of evaluating on ImageNet (Zhu et al., 2021; Tang et al., 2022; Yu et al., 2022; Chen et al., 2021). We conducted experiments on both CNN and ViT types for a comprehensive evaluation.
>
> We also have results for ResNet-18 in the paper. Even though CIFAR10 is associated with smaller models, we aimed at demonstrating the proposed method's versatility across different architectures.
>
> 5:
> We will improve the quality of tables and figures.

---

### Public Comment · ~Minsik_Cho1 · 2023-11-14
**Additional results on MobileNet-v1/v2**

* Do you have MobileNet-v1/v2 + ImageNet compression results? ResNet50 is too over-parameterized, not a good model to demonstrate the compression performance and may make a method overfitted.

* Can you also show how ResNet50 is compressed into 0.92MB with M5,B8? According to my rough estimate, it is 1.86MB.

  a) ResNet50 has ~25 million parameters.

  b) B8 replaces, each 8 parameters into a single index which is 5bit.

  c) 25e6 / 8 *5 = 15.6e6 bits ~ 1.86MB

---

> ### Author Response · Authors · 2023-11-14
>
> Thank you for your attention to this paper. Here are the responses:
>
> 1. As noted in our reply to reviewer 1, we followed the approach of other papers in the literature and focused our reporting on results for resnet18/50 and VGG + ViT. Unfortunately, we haven't conducted experiments on Mobilenet V1 or V2 so far.
>
> 2. We are unsure where exactly the 0.92MB reference you mentioned in the paper is pointing to. However, here we provide a rough estimation of our approach. In Section 3.2 of the paper, we stated, 'Unlike the quantization approach used for linear layers, here we break down the weight matrix W into B consecutive k × k 2D filters rather than individual parameters. In other words, each entry in the convolution codebook has dimensions of B×k×k.' For ResNet-50, which mainly consists of 3×3 and 1×1 2D convolution layers, we create two separate VQ codebooks for each of them. For 3×3 convolutions, if B=8, VQ essentially replaces 3×3×8 parameters with a single index vector. For 1×1 kernels, we simply replace 8 numbers with one index. Assuming one byte is required to store the index, for 3×3, we achieve a reduction factor of 1/(3×3×8×4), and for 1×1, a factor of 1/(8×4). Here we assumed one byte per index, while in the case of M=16, just 4 bits will be sufficient for each index. It's important to note that we do not apply VQ to any batch-norm or bias parameters.

---

### Meta-Review · Area_Chair_9T1W · 2023-12-11

**Metareview:**

This paper proposes to compress and accelerate neural networks with vector quantization. Reviewers agree that VQ is a reasonable and promising approach to reduce model size. Yet some major concerns remains, including practical speedup results, experimental setup, and clarity of presentation.

**Justification For Why Not Higher Score:**

lack of practical speedup results, experimental setup, and clarity of presentation

**Justification For Why Not Lower Score:**

N/A

---

### Decision · Program_Chairs · 2024-01-16

Reject